# Characteristics of Underwater Topography, Geomorphology and Sediment Source in Qinzhou Bay

**Chao Cao** [1,2,3,*] **, Feng Cai** [1,2,3] **, Hongshuai Qi** [1,2,3,4] **, Yongling Zheng** [1] **and Huiquan Lu** [1]

1　Third Institute of Oceanography, Ministry of Natural and Resources, Xiamen 361005, China;
caifeng@tio.org.cn (F.C.); qihongshuai@tio.org.cn (H.Q.); zhengyongling@tio.org.cn (Y.Z.);
luhuiquan@tio.org.cn (H.L.)
2　Fujian Provincial Key Laboratory of Marine Ecological Conservation and Restoration, Xiamen 361005, China
3　Southern Marine Science and Engineering Guangdong Laboratory (Zhuhai), Zhuhai 591000, China
4　Observation and Research Station of Coastal Wetland Ecosystem in Beibu Gulf, MNR, Beihai 536015, China
*　Correspondence: caochao@tio.org.cn; Tel.: +86-18030085312

**Abstract:** Human activities for exploitation and utilization of coastal zones have transformed coastline morphology and severely changed regional flow fields, underwater topography, and sediment distribution in the sea. In this study, single-beam bathymetry coupled with sediment sampling and analysis was carried out to ascertain submarine topography, geomorphology and sediment distribution patterns, and explore sediment provenance in Qinzhou Bay, China. The results show the following: (1) the underwater topography in Qinzhou Bay is complex and variable, with water depths in the range of 0–20 m. It can be divided into four underwater topographic zones (the central (outer Qinzhou Bay), eastern (Sanniang Bay), western (east of Fangcheng Port), and southern (outside of the bay) parts); (2) based on geomorphological features, the study area comprises four major submarine geomorphological units (i.e., tide-dominated delta, tidal sand ridge group, tidal scour troughs, and underwater slope) and two intertidal geomorphological units (i.e., tidal flat and abrasion platforms); (3) sandy sediments are widely present in Qinzhou Bay, accounting for 70% of the total sediments. From the mouth of the Maowei Sea to the central and northern part of Qinzhou Bay, the sediments gradually become coarser, shifting from sandy mud to muddy sand, and then to fine sand and medium–coarse sand, especially inside the trench. The detrital minerals contained in the sediments mainly consist of quartz, feldspar, ilmenite, leucosphenite, tourmaline, and detrital minerals, whereas the clay minerals are dominated by kaolinite, followed by illite and smectite. The sediment provenance is mainly terrigenous input from near-source river. With sea reclamation and dam construction, outer Qinzhou Bay has experienced enormous morphological variation of its coastline. Human activities for exploitation and utilization of coastal zones have transformed coastline morphology and severely changed regional flow fields, underwater topography, and sediment distribution in the sea. Together with the channel effect where the velocity of ebb tide is greater than that of flood tide, the underwater topography is characterized by increased scale and height difference of troughs and ridges as well as enhanced offshore deposition.

**Keywords:** underwater topography and geomorphology; sediment type and source; human activity; Qinzhou Bay

## 1. Introduction

As is well known, estuaries and coasts are the areas where human activities are the most frequent and the degree of development is the highest [1]. As the interaction zones of land and the ocean, these areas are not only affected by runoff, tidal currents, waves, wind, solar radiation, coastal currents, and other water and biological processes, but also frequently affected by human activities [2–7]. The morphological structure and evolution processes of topography and geomorphology are affected by geological structure, sea

level rise and fall, marine hydrodynamics, hydrochemical composition, climate, biology, and human activities [8–10]. At the same time, coastal zones have the functions of tourism and leisure, disaster prevention and mitigation, and ecological maintenance [1,11–14]. They are the core elements and important driving forces for the rapid development of the coastal tourism industry and play an indispensable role in the rapid development of coastal tourism and the construction of ecological civilization [13–15].

The interaction between land and sea brings abundant primary productivity and sediment, and forms complex submarine topography and tidal current deposits in estuary areas, which are suffering from erosion and degradation to varying degrees, leading to great pressure and challenges for the development of coastal ecological environments and tourism economies [16]. The deposition process and dynamic geomorphology in coastal zones depend on the interaction between the sediment budget and the dynamic coastal environment, and the balance of the sediment budget is mainly related to change of the supply capacity of the provenance area [17–20]. Especially in estuaries and bays, coarse-grained sandy sediments mainly come from near-source river sediment transport and land surface erosion products, whereas argillaceous sediments can be transported over long distances, even across the continental shelf and basin [21–23]. In studies on sedimentary environments and depositional processes, grain size composition, grain size parameters, and various provenance plots of sediments indicate the characteristics of the sedimentary environment and the processes of material transport [24,25]. Clay minerals can be used to trace the migration and variation of suspended matter with ocean currents, whereas detrital minerals can also trace the transport processes of sediments from source to sink; these minerals together are of great significance for elucidating the provenance of estuarine sediments [26,27].

Humans are altering the planet, including long-term global surface geologic processes, in such a way that a new geological epoch called the Anthropocene has been introduced, although still waiting to be officially recognized. While the study of land-use changes in the Anthropocene has considerably advanced, much less is known about the human impact on the seafloor, particularly in coastal areas. Inlets play a key role in the evolution of coastal barrier systems given that they: (i) control hydrodynamics, i.e., the rate of water exchange which influences the chemical–physical properties of the lagoon; (ii) are responsible for the sediment transport from the lagoon to the open sea and vice versa; (iii) affect the morphodynamics of the adjacent coast; (iv) allow the migration of different species at different life stages. In addition, tidal inlets are often subject to intense maritime traffic and human modification [28].

Qinzhou Bay is strategically located and rich in natural resources such as aquatic products, minerals, beaches, islands, and bays, which are also active places for human exploitation and utilization. In recent years, with the needs of economic development, the construction of sea reclamation facilities such as ports, docks, petrochemical industrial facilities, and nuclear power stations has dramatically changed the coastline morphology; the areas of water and tidal flats have decreased sharply, whereas the underwater topography and geomorphology have varied drastically, causing damage to fishery production and the ecological environment that will be difficult to reverse [29,30]. Therefore, for this study, the underwater topographic and geomorphologic features in outer Qinzhou Bay were investigated based on bathymetric data from the "908 Project" (National Project of Comprehensive Investigation and Research of Coastal Seas in China). The formation mechanisms underpinning the underwater topography and geomorphology in the study area were explored by analyzing the compositional characteristics of detrital minerals and clay minerals in sediments and the provenance of the sediments. The results could provide technical support for rational exploitation and utilization as well as ecological restoration and conservation in the bay.

## 2. Study Area

Qinzhou Bay is located at the top of Beibu Gulf and the middle part of the Guangxi coast, within 108°28′20″–108°45′30″ E and 21°33′20″–21°54′30″ N. The bay is composed of an inner Bay (Maowei Sea) and outer bay (Qinzhou Bay). It is a semi-enclosed natural bay with a narrow middle and wide ends. It is surrounded by land on the east, west and north, and connects with Beibu Gulf to the south. The mouth of the bay is 29 km wide and 39 km deep. The total coastline of the entire bay is 336 km, and the bay area is 380 km², of which the beach area is 200 km², with water depths of 10–20 m [31]. This study focuses on the outer bay of Qinzhou Bay, which was Qinzhou Bay for short.

The geological structure and geomorphology of Qinzhou Bay are complex, belonging to the second uplift zone of the Cathaysia system [31]. In terms of the regional structure, it belongs to the Guangpo anticline and Xiaodong-Fangcheng fold–fault zone, and the structural line direction is NE–SW (Figure 1). The faults in this area are well developed, and the NE- and NW-trending faults constitute "X"-type faults [32]. The Guangpo anticline is notably cut and displaced by NW-trending faults near Longmen, with broken rock strata and river cutting. As a result of the transgression in the late glacial period, there are numerous islands in the bay, many branches, and winding coastline, forming a drowned valley type bay. There are many islands in the east of the bay, and the flow channels around them are essentially free of sediment shoals; most of them have deep waters with rocky banks. There are fewer islands in the west of the bay than in the east, but there are numerous tidal branches, forming more small bays and shoals in this part of the bay.

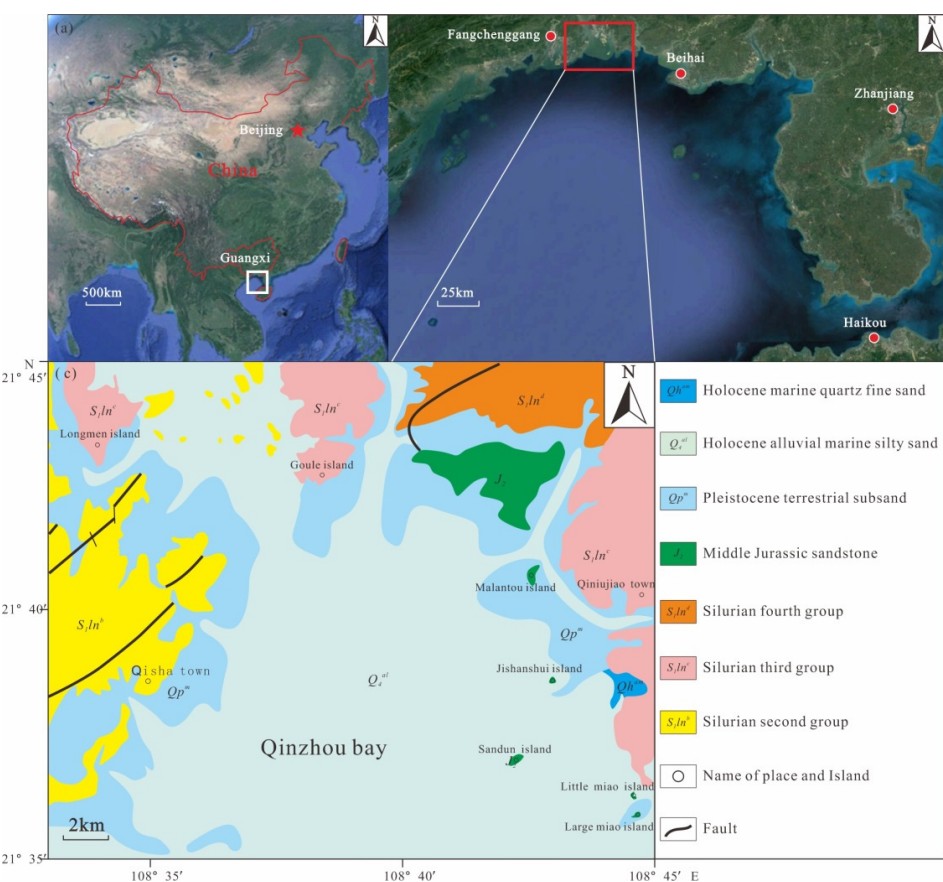

**Figure 1.** Diagram of regional structures and stratum structures in the research area. Data source Google earth map and China Gulf Chronicle [30].

In the northern part of Qinzhou Bay, the Maoling River and Qinjiang River converge. The average annual runoff of the Maoling River is $15.97 \times 10^8$ m³, and the average annual sediment discharge is $31.86 \times 10^4$ t; the average annual runoff of the Qinjiang

River is $11.69 \times 10^8$ m³, and the average annual sediment discharge is $26.99 \times 10^4$ t. Sediment carried by these two rivers has been deposited near the estuary and continuously pushed forward to the sea, forming a large sandy and muddy shoal with an area of about $1.1 \times 10^4$ hm [31], whereas the shoals on both sides of the outer bay formed an area of about $0.9 \times 10^4$ hm. At the outlet to the south of Longmen Port in the central part of the bay, tidal current deep troughs and tidal sand ridges alternate and radiate toward the sea to form a tidal delta. The abundant beach resources in the bay are very beneficial to the development of agriculture and aquaculture. To the south of Longmen Port, the tidal currents and deep channels are smooth and there is relatively little siltation, which is conducive to the development of marine transportation [31–33].

Qinzhou Bay is a gourd-shaped bay with a winding coastline, scattered islands, and complex tidal currents (Figure 2). The tides comprise regular diurnal tides and irregular diurnal tides. The movement form of the tidal currents is characterized by reciprocating flow. The long axis direction of the ellipse of the main diurnal component is basically consistent with the channel trend, which is oriented S–N. The rotation rate of the tidal current ellipse is between 0.0 and 0.5. The rotation direction of the tidal current ellipse is clockwise except for the top of the bay. The average flood velocity is 38.6–53.7 cm/s, and the average ebb current velocity is 54.8–77.2 cm/s. The regional variation of velocity is large, with larger values in the outer bay than in the inner bay, and the maximum value at the bay neck. The flow direction is mostly south or SSW. The flow velocity in the surface layer is greater than that in the bottom layer [31,34].

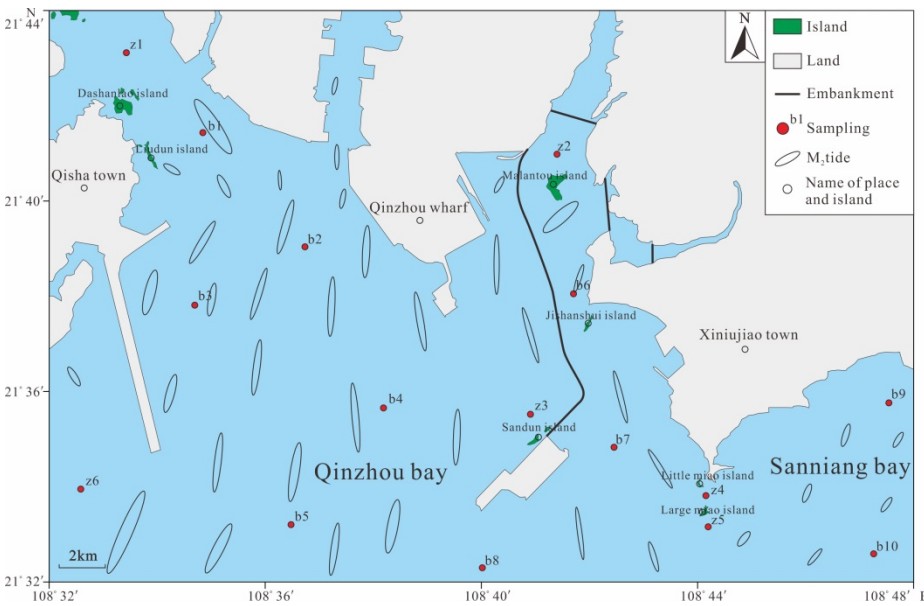

**Figure 2.** Distribution of sampling sites in the research area. [30].

The strata exposed around Qinzhou Bay are mainly Silurian (Paleozoic), Jurassic (Mesozoic), and Quaternary (Cenozoic). Holocene medium–fine sand, silty clay and silt are widely distributed, and Holocene alluvial plain, river–sea mixed accumulation delta, marine erosion platform, marine plain, tidal flat, estuary bar, tidal sand ridge, underwater delta, and underwater bank slope landforms, as well as other estuarine and coastal landforms, are generally well developed.

## 3. Materials and Methods

### 3.1. Underwater Topographic Survey and Sample Collection

In March 2017, an underwater topographic survey was carried out in the study area using an HY-1601 single-beam echosounder (Haiying, Wuxi, China). The instrument depth range is from 0.3 m to 300 m, depth accuracy is ±1 m (water depth), resolution is 1 cm, draft

adjustment range is 0.0 m to 9.0 m, operational frequency of the measurements is 208 KHz, sound velocity adjustment range is 1300 m/s to 1700 m/s. The single-beam bathymetry lines were arranged perpendicular to the coastline, with a scale of 1:5000. The bathymetry lines were spaced 50 m apart, and two to three detection lines were set perpendicular to the main survey line during a cruise of the Boat Qinyu 26957 in 2017 [35]. In total, 300 km of main survey line and 10 km of detection lines were arranged, which fully covered waters of greater than 0-m depth in the outer Qinzhou Bay study area.

The sounding rod is firmly installed on the middle side of the working vessel to accurately measure the depth of the sounding probe into the water. The receiving antenna of the positioning instrument and the transducer of the sounding instrument are installed on the same plumb line, which eliminates the position difference between the locating point and the sounding point [36,37]. The instrument was carefully adjusted, strictly compared, and checked before operation every day, including proofreading atlas and digital signals, measurement of sound velocity and measurement of water depth on the comparison panel, etc. The difference between the recorded water depth and the depth measured by the instrument was controlled within 5 cm. The sound velocity is measured by RBRconcerto CTD, which has a depth of 100 m and a temperature range of $-5\,°C$ to $+35\,°C$. The measurement accuracy of sound velocity can reach $\pm0.1$ m/s, the resolution is 0.05 m/s, the measurement range of sound velocity is 1400–1600 m/s, and the measurement speed is 30 Hz. During the sounding operation, the operation is strictly in accordance with the instrument operation instructions, especially paying attention to shift gears in time to ensure the integrity of the map. DGPS positioning system is used for offshore operation positioning, and the sailing positioning accuracy is better than 1.0 m. During the operation, the Hypack software synchronizes the marking and automatically records the depth and location data, which are stored in the navigation system.

Sixteen sediment samples were collected from different geomorphological units, including tidal creeks, sand ridges, deep troughs, abrasion platforms, and underwater slopes, in March 2017. We grabbed to obtain surface sediment samples and collected columnar sediment samples with gravity tube. The sediment samples comprised ten surface samples and six core samples (15–70 cm in length). The surface samples were assigned with one sample per station, and the core samples were divided with one sample every 5–10 cm, giving a total of 31 samples. In-situ measurements were performed to determine sediment parameters such as moisture content and redox potential (Eh). The samples were sealed in PVC tubes, labeled, and refrigerated. Sediment grain size and the compositions of detrital and clay minerals were analyzed in the laboratory. The specific sampling points and sample information are shown in Figure 2 and Table 1.

**Table 1.** Data of sediments collected in the research area.

| No. | Longitude | Latitude | Length of Core/cm | Water Depth/m | Sediment Type |
|---|---|---|---|---|---|
| z1 | 108°33′52″ | 21°43′26″ | 32 | 3 | Clay sand |
| z2 | 108°41′48″ | 21°41′32″ | 15 | 1 | Fine sand |
| z3 | 108°48′50″ | 21°35′42″ | 48 | 2 | Silty sand |
| z4 | 108°44′32″ | 21°35′45″ | 55 | 2 | Fine sand |
| z5 | 108°44′33″ | 21°35′14″ | 26 | 2 | Fine sand |
| z6 | 108°33′21″ | 21°35′34″ | 70 | 5 | Clay sand |
| b1 | 108°41′46″ | 21°35′17″ | — | 10 | Silty sand |
| b2 | 108°37′4″ | 21°39′35″ | — | 6 | Fine sand |
| b3 | 108°35′5″ | 21°38′46″ | — | 4 | Clay sand |
| b4 | 108°38′46″ | 21°37′22″ | — | 7 | Silty sand |
| b5 | 108°37′37″ | 21°34′57″ | — | 8 | Sandy silt |
| b6 | 108°42′12″ | 21°39′1″ | — | 1 | Clay sand |
| b7 | 108°42′48″ | 21°36′42″ | — | 4 | Clay sand |
| b8 | 108°40′1″ | 21°34′16″ | — | 6 | Clay sand |
| b9 | 108°45′21″ | 21°45′29″ | — | 1 | Fine sand |
| b10 | 108°45′38″ | 21°36′45″ | — | 3 | Sandy silt |

### 3.2. Bathymetric Data Processing and Sample Analysis

As required by the *Technical Regulations for Surveying Seabed Topography and Geomorphology* of the National Project of Comprehensive Investigation and Research of Coastal Seas in China [38], fusion of bathymetric data and tidal level data was processed using the Hypack software (v18.1, Hypack Inc., New York, NY, USA). The continuity and confidence degree of navigation positioning data and single-beam data were examined through software identification and manual interpretation. Combining tidal level correction and data fusion, effective bathymetric data and ASCII code files were generated.

ArcGIS (v9.1) was used to evenly select 4–5 coordinate control points on the digital chart for geographic registration, with an average mean square error of 1.9 m [39,40]. Kriging interpolation was used to interpolate 50 m × 50 m grid spacing of water depth data after tidal level correction. During the bathymetric survey, the GXCORS non-tidal station was used to synchronously calibrate the tide level. In this study, the planar control solution of 5 control points in the test area was carried out by using GAMIT software and 5 IGS stations (BJFS, Hyde, Pimo, Shao, TWTF). The ArcScan tool was used to automatically extract isobaths, coastline, coastal structures, and other elements. Man–machine interaction was used to proofread water depth data [41]. The instrument displays the real-time seabed depth data graphically, identifies the acquired depth data, and provides qualified depth profile lines for manual evaluation. The processing personnel can interpret and correct the false data in real time. Finally, a model of the underwater digital topography of Qinzhou Bay was generated by Fledermaus v7.3 (IVS 3D Inc., Fredericton, NB, Canada) (Figure 3).

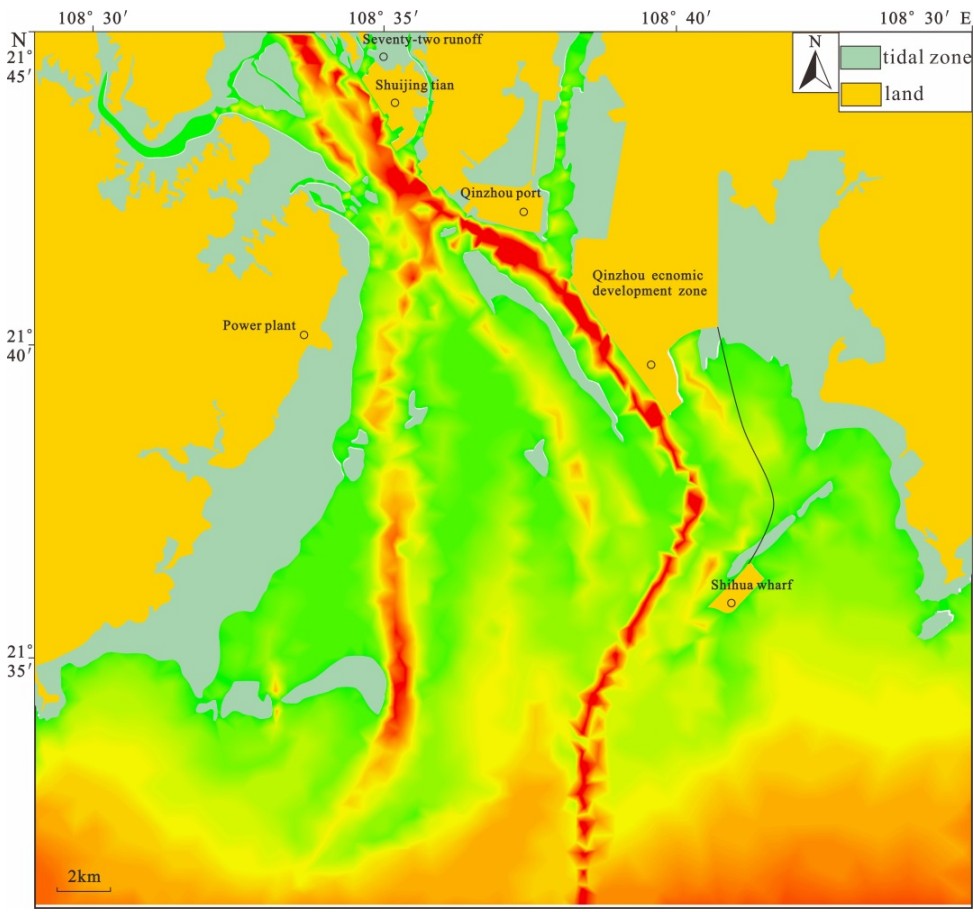

**Figure 3.** Bathymetric data and water depth in the research area.

The grain-size distribution of bulk terrigenous sediments (carbonate- and organic-free) was analyzed using a Beckman Coulter LS-230 (Series 230) laser particle size analyzer. The preparation and pretreatments of the sediment samples followed the method described in

detail by Zhao and Liu (2018) [15]. Briefly, any organic matter in bulk sediment samples was first removed using 30% $H_2O_2$ (3 h) in a water bath at 60 °C. The samples were then decarbonized via complete reaction with 0.5% HCl using a magnetic stirring apparatus (60 min). The acidified samples were then washed with distilled water several times until the pH ≈ 7. Binocular observation of selected samples showed that the proportion of opal was quite limited, implying that opal had little influence on the terrigenous grain sizes of the three studied cores. All samples were measured twice and showed good repeatability. The mean grain-size (MGS) was obtained following the method proposed by McManus (1988) [42], while the sensitive grain-size components were identified by plotting various grain-size classes versus their standard deviations [43]. A higher standard deviation normally represents a population of terrigenous particles with higher variability in response to sediment dynamic changes. Finally, the MGS of each sensitive grain-size component was calculated according to the method of Xiang et al. (2006) [44]. The test was carried out with Coulter LS-230 laser particle size analyzer. The mean grain size (Mz), standard deviation (σI), skewness (SKI), and kurkity (KG) were calculated by Sheppard particle size calculation formula. The grain size analysis was completed in the Third Institute of Oceanography, Ministry of Natural Resources.

During the separation of sediment minerals, samples of about 50 g were selected for wet screening, and 0.063–0.125 mm samples were selected for drying. Tribromomethane (specific gravity 2.89 g/cm$^3$) was used to separate heavy and light minerals [45]. The separated samples were rinsed repeatedly with alcohol, dried at 60 °C and weighed. Among them, the light minerals were made into mineral sheets with a mixture of epoxy resin and triethanolamine at a ratio of 6:1 and baked at a constant temperature of 70 °C for 48 h. Three to five microscopic visual fields were selected for mineral identification based on the light mineral thin sections, and the average value of each visual field was taken to reduce the analysis error. Under the solid microscope and polarized light microscope, the number of particles identified in each heavy mineral sample was 400, and the percentage of each heavy mineral was calculated.

Clay minerals were identified by X-ray diffraction (XRD) of oriented mounts of decarbonated clay-sized (<2 m) particles using a PANalytical X'Pert PRO diffractometer. Every sample was subjected to three XRD runs under the conditions of air-drying, ethylene-glycol solvation (>24 h), and heating (490 °C, 2 h). Then, three XRD diagrams were obtained and used to identify and interpret clay minerals, according to the (001) basal reflections. The peak areas of smectite (15–17 Å), illite (10 Å), and kaolinite/chlorite (7 Å) on the glycolated curve were calculated semi quantitatively using the MacDiff software (Petschick, 2000) [46]. The ratio of the 3.57/3.54 Å peak areas was used to determine the relative proportions of kaolinite and chlorite. In particular, the weighting factors proposed by Biscaye (1965) [47] were not used when calculating the relative weight percentages of each individual clay mineral [48–50]. A precision of ±2% (2σ) for the XRD method was inferred from replicate analyses of selected samples, and an accuracy of ~5% was determined for the semi quantitative evaluation of each clay mineral species. Additionally, illite crystallinity was inferred from the half height width of the 10 Å peak on the glycolated curve, whereas the illite chemical index was inferred from a ratio of the 5 Å and 10 Å peak areas.

## 4. Results

### 4.1. Zoning of Underwater Topography and Associated Features

The present study area is outer Qinzhou Bay, which includes the water area of Qinzhou Bay except the Maowei Sea. According to major factors, such as coastline morphology, isobath distribution, and underwater topographic relief, the underwater topography of the study area can be divided into four representative zones. These are the underwater topographic zone of outer Qinzhou Bay in the central part, the underwater topographic zone of Sanniang Bay in the eastern part, the underwater topographic zone east of Fangcheng Port in the western part, and the underwater topographic zone outside of the bay in the southern part (Figure 4). The main features of these zones are as follows.

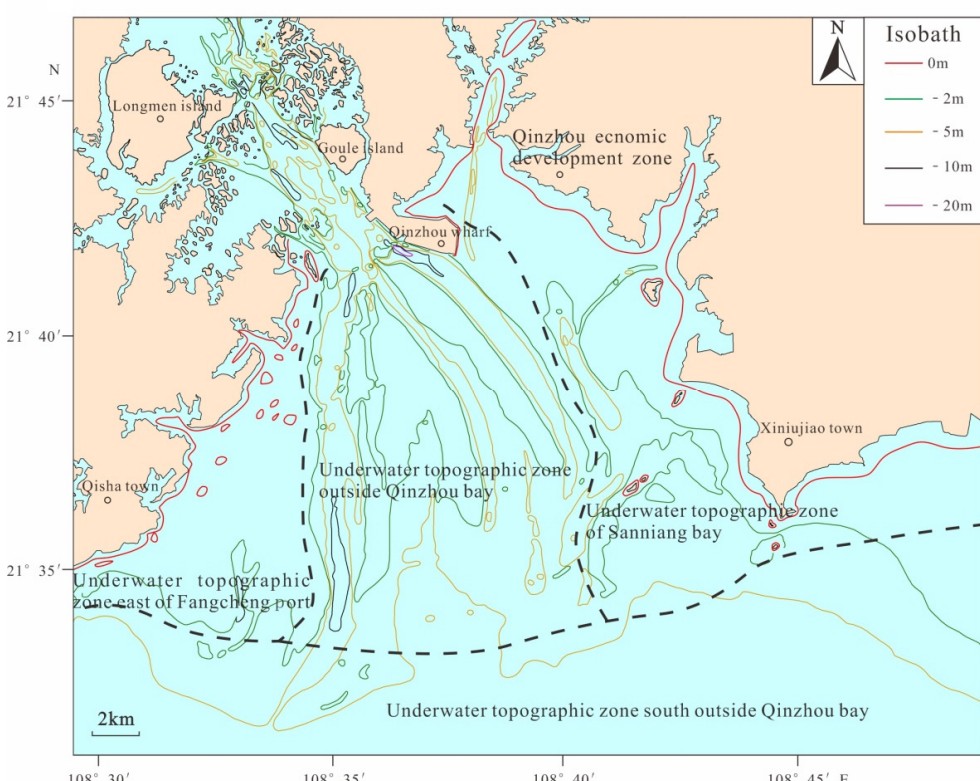

**Figure 4.** Submarine topography in the research area.

In the central part, the underwater topographic zone of outer Qinzhou Bay is the core region of the water area in this study. It is an estuarine sea facing the Maowei Sea, with shallow water depths generally in the range of 0–5 m. Considered the overall trend of the topography, the water depths are relatively shallow in the near-shore waters to the north, and gradually increase with increasing distance to the shore; the overall topography is very flat. The isobaths in the bay as a whole protrude from northwest to southeast, and an underwater shoal extends toward the sea in a duck tongue shape. Similarly, two remarkable areas with relatively dense isobaths can be seen in the central part of Sanniang Bay. The water depths in these two areas are greater than those in the adjacent areas of the same latitude, forming two troughs in the underwater shoal. The trenches are roughly SN and NE–SW. One trench is ~4 km long and 500 m wide with a water depth of 4 m; the other is ~6 km long and 700 m wide with a water depth of 4.5–5.5 m. Together with the channel effect where the velocity of the ebb tide is greater than that of the flood tide, the underwater topography in this zone shows increased size and height differences of troughs and ridges, as well as enhanced deposition in the inner bay and near-shore area [51].

In the eastern part, the underwater topographic zone of Sanniang Bay varies in water depth between 5 and 20 m. Generally, the isobaths protrude from northwest to southeast. The distribution of the isobaths is relatively uniform, indicating slow gradual variation in the near-shore submarine topography. To the southeast, there is an isobath fluctuation area with water depths varying between 12 and 14 m; in this area, several isobaths show ups and downs in a banded pattern from south to north. The fluctuation of the depth ranges from 1.0 to 2.0 m, and the length of the band is 1.5 to 3.0 km, which are characteristic of a typical submarine active sand ridge. Additionally, to the west of Sanniang Bay and east of Damiaodun, there is a shallow pit area with relatively dense and closed isobaths; this area has a diameter of ~2.5 km and water depths of 5–8 m.

In the western part, the underwater topographic zone east of Fangcheng Port has a water depth of no more than 20 m. The submarine topography is relatively gentle, with local distribution of islands and reefs. The general trend of the topography is a gradual

incline from land to sea, and the isobaths are uniformly arranged along the shore. In this zone, the underwater slope is relatively wide with alternating shoals and trenches; the underwater topography is therefore relatively complex. There are natural and artificially excavated channels on the slopes of Fangcheng Port and outside of Qinzhou Bay [52].

In the southern part, the underwater topographic zone outside of the bay is a central submarine plain with isobaths of greater than 20 m. With regard to the overall trend of the topography in this zone, water depths gradually increase from north to south, and the submarine topography slowly inclines seaward. The overall topography is very flat, with a mean drop of only 2–4‰ in the slope gradient. In some areas there is a sharp increase in slope gradient, whereas in other submarine areas the topographic variation is minor. The isobaths within this zone generally protrude from northeast to southwest, and they are distributed uniformly, suggesting slow gradual change of the near-shore submarine topography, which gradually shifts from a submarine plain to a relatively steep underwater slope area.

### 4.2. Division of Geomorphological Units and Their Features

The classification of submarine geomorphology is usually based on a combination of morphology and formation mechanisms. According to this principle, the submarine geomorphology can be divided into four levels based on the distribution pattern from macroscopic to microscopic, and from groups to individuals [53,54].

Following the above-mentioned principle, the seafloor of outer Qinzhou Bay is divided into four typical level-four geomorphological units: the tide-dominated delta, tidal sand ridge group, tidal scour troughs, and underwater slope (Figure 5).

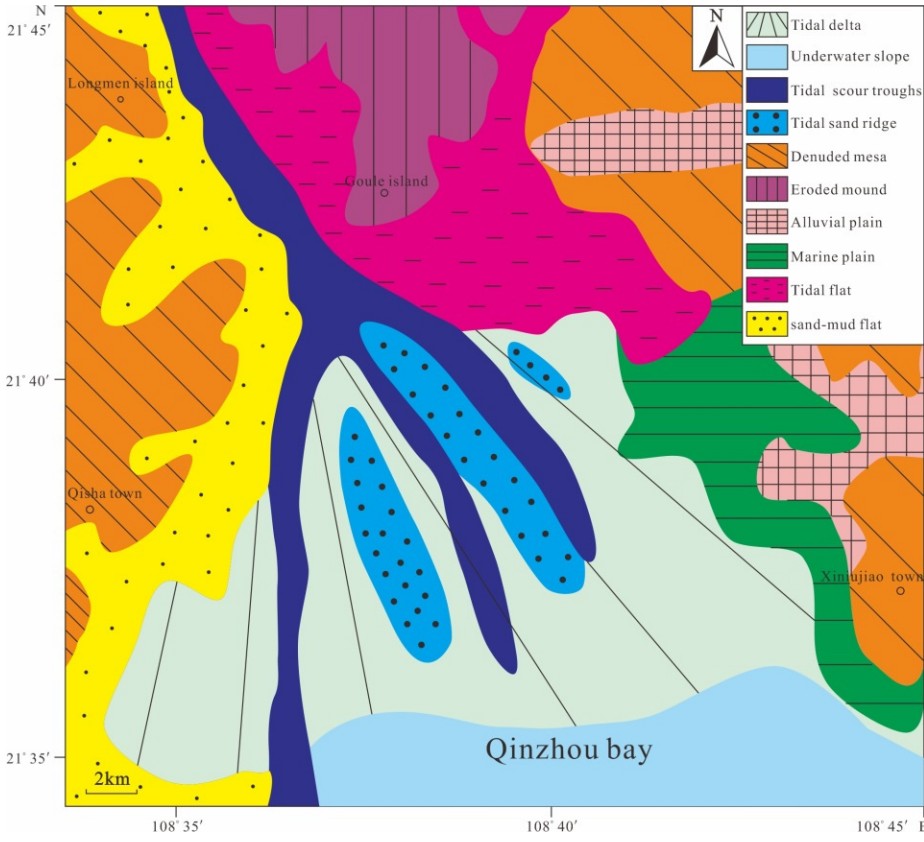

**Figure 5.** Coastal zone and submarine geomorphology in the research area [31].

The tide-dominated delta, that is, the underwater delta controlled by tidal currents, is an accumulational landform formed by flocculation of suspended colloids driven by the deposition of sediments and the mixing of freshwater with seawater, as the carrying

capacity of the river declines abruptly after the sediments it carries are discharged into the sea [54,55]. The tide-dominated delta in Qinzhou Bay is formed as follows: after the Qinjiang and Maoling rivers flow into the Maowei Sea, the materials are brought from the Maowei Sea to the outside of the Longmen Channel under hydrodynamic conditions where the ebb current velocity is greater than the flood current velocity. Because of a sudden widening of the channel, terrigenous materials are accumulated on the south side of the Longmen Channel. The sedimentary layers of this delta are relatively complex, and consist of yellow sandy mud, fine sand, and gray-black silt containing a large amount of bioclastics, gradually shifting to medium–coarse sand or gravel deposits seaward.

The tidal sand ridge group comprises ridge-shaped landforms developed in sandy sediments, with widths of tens of meters to several kilometers, and lengths of several kilometers to tens of kilometers. Because sand ridges are often distributed in a clustered pattern, they are referred to as the tidal sand ridge group [55]. In the present study area, the tidal sand ridge group is mainly developed on a large scale at the mouth of Qinzhou Bay. It spreads to the south in a radial pattern, with alternating ridges and troughs. Generally, the ridges trend NNW–SSE. Among them, Laorensha is the largest one, at ~7.5 km in length and ~0.7 km in width. The sand body trends NNW, with smaller tidal sand ridges developed on both sides. The materials comprise fine sand or clayey sand, with low silt content and moderate sorting.

Tidal scour troughs are developed in narrow channels and are characterized by long strip-shaped negative landforms formed by tidal erosion. In the study area, tidal scour troughs occur in the trenches adjacent to tidal sand ridges, and are formed by scouring of the tide-dominated delta by ebb currents with strong dynamic force. The troughs extend from the Longmen Channel (a deep trough) along the trenches of tidal sand ridges, and reach the underwater slope outside of the bay. The largest scour trough is ~15 km long, and its distribution is consistent with that of the tidal sand ridge group. The materials in the troughs are irregular in grain size and poorly sorted [56].

The underwater slope refers to an inclined slope extending from the low-water line to the edge of the continental shelf plain. The slope gradient can vary substantially, from 3′ to 11°; the width can range from 3 to 5 km to tens of kilometers. The slope gradient is often inversely proportional to the slope width. In the study area, the width of the underwater slope is relatively narrow, ranging from 0.6 to 1.0 km. The gradient of the near-shore underwater slope is relatively steep, generally in the range of 0.2‰ to 1.0‰, whereas the gradient of the far-shore underwater slope is gentle, generally in the range of 0.1–0.4‰. The sediments comprise fine sand that becomes muddy seaward. Because large amounts of estuarine materials are accumulated on the tidal sand ridge, overall, it appears to be a flat accumulational landform of the tide-dominated delta. The topography offshore of Qinzhou Bay is also relatively flat. This underwater slope is a transitional geomorphological unit between the tide-dominated delta and the continental shelf plain, which evolves on a small scale.

In addition to the four submarine geomorphological types described above, intertidal geomorphological types such as tidal flats and abrasion platforms occur in the near-shore area of outer Qinzhou Bay. The tidal flats mainly occur along the coast around Legoudou. According to their sediment characteristics, the tidal flats can be divided into low, medium, and high tidal flats. From the low to high tidal flats, the sediments change from coarse to fine, the silt content increases, and sorting changes from good to poor. Abrasion platforms occur in Longmen, Xiniujiao, and sporadic bedrock islands, with lithologic compositions of Silurian argillaceous sandstone, sandy shale, and phyllite [57–59].

*4.3. Sediment Types*

The surface sediments from Qinzhou Bay are mainly classified into the following six classes according to Shepard's classification: medium sand, fine sand, clayey sand, silty sand, sandy silt, and sand-silt-clay [60]. Sandy sediments are widely present in the study area, accounting for 70% of the total sediments. In contrast, the distributions of sandy mud and silty mud are relatively limited. The sandy sediments are distributed in strips or

patches, gradually becoming coarser from the top of the bay to the sea. The sand bodies in the outer bay show a typical distribution of tidal finger-shaped sand ridges (Figure 6). The sediments differ substantially between the east and west sides of the bay mouth area. On the east side, sediments containing coarse grains are widely present around the islands. On the west side, the sediments are fine, mainly consisting of sandy silt and muddy sand; the sand bodies are exposed at low tide, and the aggradational coast grows quickly, with multi-row dikes along the shore [61].

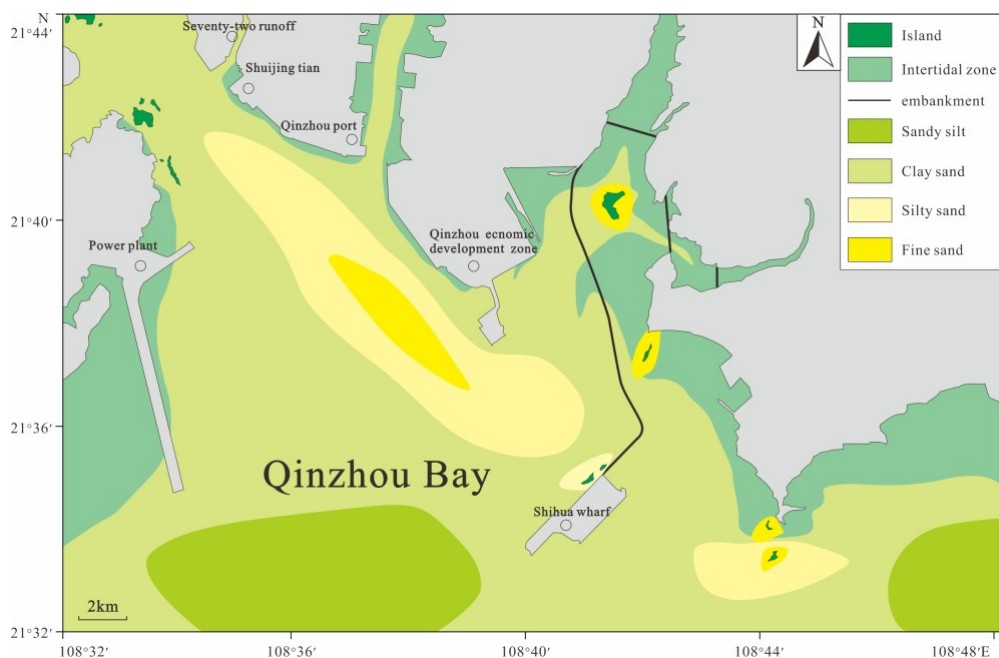

**Figure 6.** Distribution of sediment types in the research area.

The light minerals in the sediments from Qinzhou Bay are mainly composed of quartz, feldspar, and detrital minerals. The quartz is mostly sub-angular with a content of 75–85% (Table 2). The color of the quartz is light isabelline, maroon, or milky white. The tidal flat sediments mainly contain light isabelline and maroon quartz, whereas the outer bay sediments mainly contain light green, yellow, and shallow-colored quartz. The feldspar is mostly columnar with contents of 11–17%. The color of the feldspar is light isabelline and light brown in most cases. The detrital minerals primarily consist of quartz and feldspar, with its content varying substantially. The detrital minerals content is relatively high in near-shore reefs and tidal channels (~15%), whereas it is extremely low in sediments of the sand bodies on both sides of the bay mouth.

**Table 2.** Comparison of light mineral contents in the surface sediments between outer Qinzhou Bay and the Qinjiang River.

| Source | Percent by Volume/% | | | | Quantity |
|---|---|---|---|---|---|
| | **Quartz** | **Potash Feldspar** | **Albite** | **Rock Detrital Minerals** | |
| Research zone (range) | 75–85 | 4–7 | 7–10 | 3–15 | 16 |
| Research zone (average value) | 83.10 | 4.27 | 7.35 | 5.28 | 16 |
| Qinjiang river (average value) | 82.01 | 7.18 | 6.69 | 4.12 | 5 |

The heavy minerals are dominated by ilmenite, leucosphenite, and tourmaline, with contents of ~52%, ~18%, and 11%, respectively; the contents of the remaining heavy minerals are relatively low (Figure 7). The distribution of the heavy minerals is consistent with that of the sand bodies. Both the light and heavy minerals appear to be similar to the mineral assemblages found in the source areas of the Qinjiang and Maoling rivers.

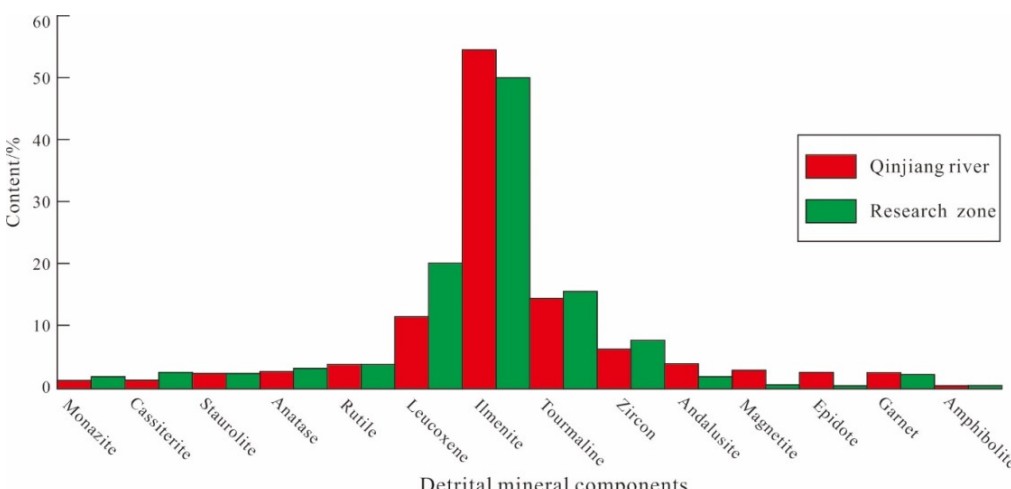

**Figure 7.** Comparison of terrestrial heavy mineral contents of surface sediments between outer Qinzhou Bay and the Qinjiang River.

## 5. Discussion

Geotectonics and neotectonic movements determine and influence the modern sea–land pattern, coastal contour, and coastal type. Geological tectonics not only controls the overall pattern and distribution characteristics of the seabed topography but also influences the hydrodynamic conditions and sedimentary environment of the modern seabed [1,62–66]. Deltaic and plain coasts were developed in the neotectonic decline area, which received thicker Quaternary ocean–land cross sediments. Not only have these coastal areas near the mouths of rivers all developed underwater deltas to varying degrees, but the abundant terrigenous sedimentary material in the shallow water has also promoted accumulation, leading to the formation of plain geomorphology [67–70]. Surface runoff carrying terrigenous material in estuaries will form high concentrations of sediment, and under the influence of tides and tidal currents, sediment in suitable locations will form estuary deposits in the bay areas with typical geomorphological features such as tidal flat, delta, and tidal sand ridge deposits [71–74].

### 5.1. Genesis Analysis of Landforms and Geomorphology

The coastal rivers that flow into the sea in Guangxi mainly include the Nanliu River, Fengfeng River, Qinjiang River, Maoling River, Fangcheng River, and Beilun River. The total annual runoff is $182 \times 10^8$ m$^3$, and of these rivers, the Nanliu River contributes the most to this amount, accounting for about 37.5% of the runoff of these six main rivers into the sea. The Nanliu River is one of the three major rivers that flow into Beibu Gulf, followed by the Qinjiang River and Fengfeng River [31]. Because the runoff of coastal rivers in Guangxi is much smaller than that of the Yangtze River and Pearl River in China, the incoming sand in the study area is not easily transported outward and deposited at the entrance due to the support of tidal currents, and only a small amount of suspended sediment is transported to the distant sea along with the ebb current and nearshore current. Land-based sediment is confined to the area near the estuary and has a small diffusion range. Large amounts of silty clay deposits are distributed in the water depth range of 10–20 m in the study area, and play an important role in the topographic and geomorphic transformation of the study area.

Beibu Gulf is an area often affected by typhoons. The formation of coastal landforms is greatly influenced by storm surges and tsunamis formed under severe weather conditions. The storm conditions caused by typhoons increase the water level, which greatly enhances the action range and impact of waves on the coastal zone, speeds up the erosion, accumulation and transport of sediment materials, and notably enhances the geomorphologic form of the study area. Northerly wind prevails in winter, and southerly wind prevails in

summer. Waves are considerably affected by the monsoon, and the strong monsoon winds lift waves strongly in the shallow sea area. Wind waves can affect the bottom layer. Under the joint action of wind waves and tides, the distribution of sand bodies in the bottom layer has been reformed to form the modern geomorphic features [32–34,75].

The coast of Qinzhou Bay is mainly silty, and has the common characteristics of bedrock, sandy, and muddy shoreline areas; thus, the whole length of the shore is scoured and silted. The coast of the gulf is a medium–strong tidal-type coast, with the characteristic of a high tidal current velocity. At the mouth of the bay, the ebb current velocity is greater than the flow velocity; for example, at the bay neck Longmen, the falling tide speed is up to 138 cm/s. Caused by tidal action, radial tidal sand ridges are formed in the outer bay. Among them, the old sand tide ridge is 7.5 km long and 700 m wide. The bay mouth area is subject to strong wave action, such that waves and sand bodies form on the east and west sides of the bay mouth. The movement of tides causes the supply of material to diverge on the east and west sides. Under the action of the Coriolis force, the main flow of high tide enters from the east of the inlet, and because its sediment content is low, it can only start the input of fine material in the east of the inlet and bring it into the inner inlet during this movement. The ebb tide also brings material input from the river in the inner bay; thus, its sediment content is high and a considerable amount of material can accumulate on the west side of the bay mouth. Therefore, west of the mouth of the bay, there are broad underwater shoals, such as sand. From Shanxin Village to the lower slope, there are four to five rows of sand dykes, forming a coastal sand body up to 1 km wide, reflecting the continuous accumulation of the coastal deposits. At the same time, the sandy intertidal zone is also broad, with a width of about 500 to 1000 m, and the beach becomes stacked. The east coast of the mouth of the bay is narrow, and the shoreline is stable or subject to weak erosion. Because the Longmen waterway is narrow and the outer bay is open, the ebb tide transports Maowei Sea materials and spreads them along the channel to the outer bay. Therefore, a complex geomorphic unit with tidal sand ridges and tidal scour grooves is developed in the outer bay of Qinzhou Bay.

*5.2. Sediment Sources*

Coastal sediments are transported from the middle and upper rivers to the estuaries, coast, bays, and underwater areas. The compositions of sediment detrital minerals and clay minerals exhibit notable spatial heterogeneity; heavy mineral and kaolinite contents are greatly reduced, whereas light mineral, illite, and montmorillonite contents show relative increases.

Qinzhou Bay is shaped like a calabash bay with a narrow neck and different hydrodynamic conditions in the inner and outer bays; consequently, its filling and coastal scouring and silting show notable differences. Filling mainly occurs in the inner bay and all of the submerged bays, whereas filling in the outer bay is weak and scouring in the bay neck is apparent.

In Qinzhou Bay, the distribution area of sandy sediment is wide, accounting for 68% of the total area, and the distribution of sandy mud and silty mud is limited. From the top of the bay to the sea, the sediments gradually coarsen, from sandy mud to argillaceous sand, and then to fine sand and medium coarse sand. There are coarse sand, very fine sand, and very coarse sand in the middle of the bay neck, and even the bedrock is exposed at the bottom of the bay. In Lujiao Bay, on both sides of the bay neck, mud is often covered in the parts with breccia distribution. The outer bay sand bodies are distributed along the tidal current direction in strip shapes, which is typical for tidal sand ridges. There are often long strips of argillaceous distribution areas between tidal sand ridges. The difference of sediment deposits between the east and west sides of the bay mouth area are very apparent. In the east, the bedrock is exposed, gravelly coarse sand is widely distributed, and the coast is eroded backward; the sediment on the west side of the bay mouth is fine, mainly composed of fine sand and medium fine sand. At low tide, the sand body often emerges

from the water, the coast rapidly undergoes silt deposition, and there are many breakwaters along the coast.

The clay minerals in Qinzhou Bay are mainly kaolinite, followed by illite and montmorillonite. The contents of these three clay minerals vary regularly from north to south. The kaolinite content gradually decreases and the contents of illite and montmorillonite gradually increase, but the content of montmorillonite increases relatively little. The change of montmorillonite is positively correlated with the chlorine content of the seawater, but the sedimentation rates of different clay minerals in seawater of different salinity also differ. The three end-member provenance indicator map of clay minerals (Figure 8) shows that the clay minerals in the sediments of Qinzhou Bay are quite different from those of the major rivers in China; thus, long-distance input of soil dust brought by monsoon winds is not likely. Therefore, clay minerals in this area are mainly from nearshore rivers (similar to detrital minerals) and supplemented by land–sea mixed materials brought by tidal currents [75–77].

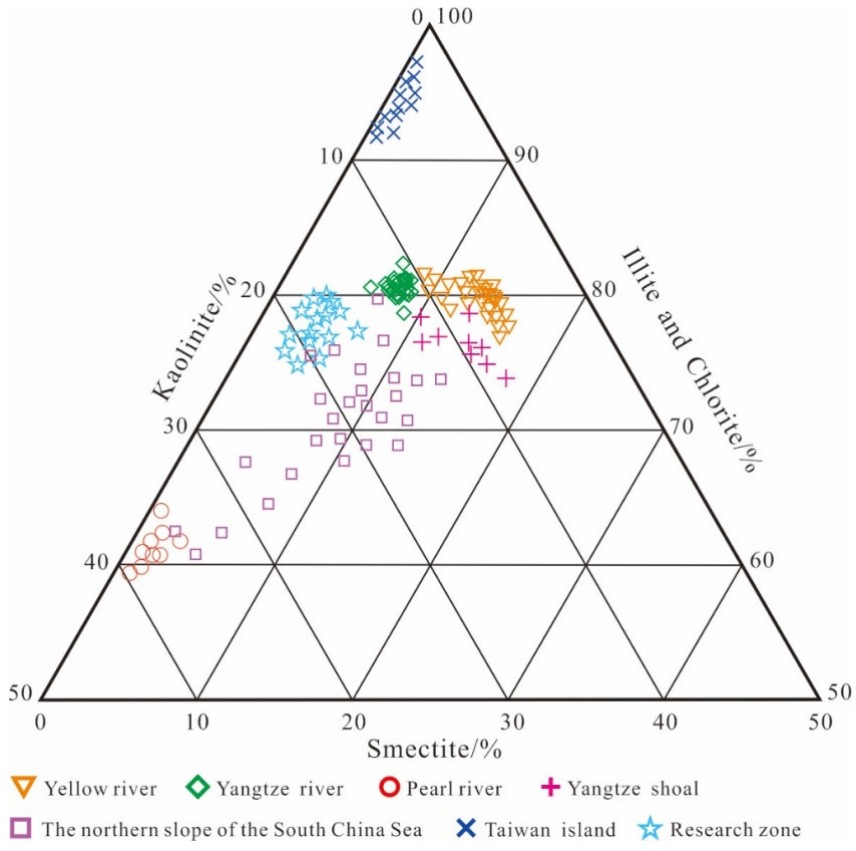

**Figure 8.** Diagram indicating the contents of three major clay minerals in the sampled sediments.

*5.3. Influence of Human Activities*

Human activities have great impacts on the development of estuaries and bays. With economic, scientific, and technological development, the exploration and understanding of submarine topography and geomorphology have further improved. While exploring and utilizing the ocean, human beings also participate in the evolution process of submarine topography [78–80]. The excavation and dredging of artificial waterways, construction of wharves, artificial reclamation, and offshore oil exploitation not only directly change the original appearance of the seabed, but also change the related marine hydrodynamic conditions, breaking the balance of scouring and silting of the seabed, and thus shaping new geomorphic forms. Tidal currents play a large role in the erosion, transport, and deposition of sediments in shallow water environment. The distribution and depth of seafloor scouring suggests that tidal flow is locally intensified by coastal geometry and



bathymetric barriers, resulting in concentrated scouring where tidal flow is restricted or redirected. In addition, superimposed bedforms reflect localized variations in flow direction that have likely developed across a range of spatial and temporal scales [28].

In recent years, the rapid decrease of river sediment supply caused by high-intensity human activities has led to a significant reduction in the sources of beach sand. The beaches and underwater deltas in the study area have been degraded to varying degrees, and the width of the sandy sediment area has been reduced. The stability of coastal geomorphic bodies depends on the sediment budget and changes to the dynamic environment, which are the result of the long-term interactions between them. However, because of the influence of human activities, such as reservoir dam construction, river sediment exploitation, soil and water conservation, and bank consolidation, the sediment input from the river entering the sea in the study area has decreased sharply since the 1990s, and notable erosion has occurred in the estuary delta and adjacent nearshore beach because of the imbalance of the sediment budget and dynamic conditions [81–85].

In the past 30 years, with increasing intensity of exploitation and utilization in Qinzhou Bay, dramatic morphological variation has occurred in the coastline and underwater topography. As shown in Figure 9, in the late 1980s, the surrounding areas of Qinzhou Bay had not yet been exploited or utilized; therefore, the coastline of Qinzhou Bay was basically dominated by natural coastline. The Maowei Sea and outer Qinzhou Bay had wide channels and smooth water flow, which maintained the topographic and geomorphological features of the natural and original coastal zone. In the mid-1990s, with the construction of Qinzhou Port, sea reclamation with water wellfields commenced; the inner bay in the eastern part of Qishierjing gradually transformed to land, while the tidal flat area shrank. In 2005, Qinzhou Port gradually expanded outward, with its southern edge extending into Qinzhou Bay. Meanwhile, sea reclamation led to a narrowing of the water channel on the east side, forming a reclaimed area of nearly 20 km$^2$. In this stage, sea reclamation for the Qinzhou Economic Development Zone on the east side began to take shape. In 2008, the construction of the Qinzhou Economic Development Zone was completed and the zone came into service; the reclaimed area reached 42 km$^2$, and the tidal flat area of Qinzhou Bay shrank further. As of 2012, construction of the Sandun Island Petrochemical Dock was completed, which only formed an effective reclamation area of 2.3 km$^2$, but also resulted in a ~10-km-long transport connection embankment; this embankment hindered the freshwater input of runoff in the eastern part of Qinzhou Bay and seawater exchange at the bay mouth, which severely changed the regional hydrodynamic conditions and sediment transport status. In 2019, with the construction of Qinzhou Nuclear Power Station in Hongxing Village, Qisha Town, not only was a reclaimed area of 3 km$^2$ formed, but two water intake protection embankments with lengths of nearly 7 km were also built in the south and north, respectively; these factors altered the hydrodynamic conditions and sediment transport and redistribution patterns in the western part of Qinzhou Bay. Thus far, human activities over the past 30 years have resulted in a newly added land area of nearly 70 km$^2$ and artificial structures ~20 km in length, while the tidal flat area has been reduced to 320 km$^2$. These changes could account for the present morphologic features of Qinzhou Bay, with a staggered distribution of artificial coast, islands and reefs, and natural coastline coupled with complex underwater topography and geomorphology.

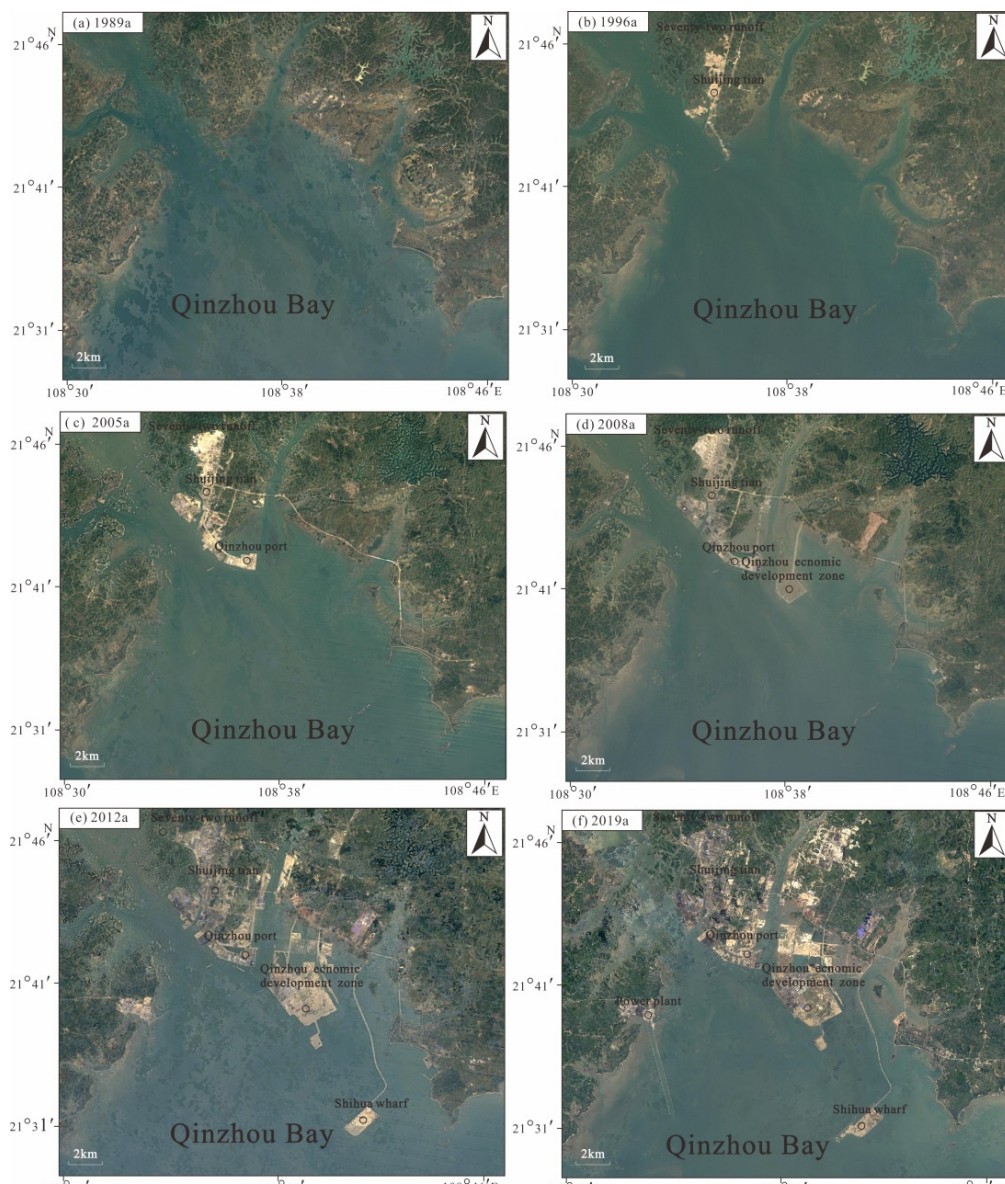

**Figure 9.** Multi-phase images of sea reclamation in outer Qinzhou Bay in a recent 30-year interval.

## 6. Conclusions

The underwater topography of outer Qinzhou Bay is complex and variable, with water depths of 0–20 m. It can be divided into four underwater topographic zones in the central (outer Qinzhou Bay), eastern (Sanniang Bay), western (east of Fangcheng Port), and southern (outside of the bay) parts, respectively. The geomorphology can be divided into four primary submarine units (i.e., the tide-dominated delta, tidal sand ridge group, tidal scour troughs, and underwater slope), and two secondary intertidal units (i.e., the tidal flat and abrasion platforms). Driven by the force of ebb currents, the materials discharged into the sea by the Qinjiang and Maoling rivers are subjected to the channel effect of the Longmen Channel and sediment flocculation because of freshwater–seawater mixing. Consequently, this area evolves composite geomorphological units of the tide-dominated delta associated with the tidal sand ridge group and tidal scour troughs.

Sandy sediments occur widely in outer Qinzhou Bay, accounting for ~70% of the total sediments. The sediments gradually become coarser from the top of the bay to the outer bay. The sand bodies in the outer bay are distributed in a strip-shaped pattern along the tide direction, and thus are typical tidal sand ridges. Sediment detrital minerals mainly

include quartz, feldspar, detrital minerals, ilmenite, leucosphenite, and tourmaline. The clay minerals are dominated by kaolinite, followed by illite and smectite. The sediment provenance is primarily terrigenous input by near-source river, supplemented with marine input by tides.

With sea reclamation and dam construction, the coastline morphology of Qinzhou Bay has varied dramatically. Human activities over the past 30 years have formed a new land area of nearly 70 km² and artificial structures of ~20 km in length; meanwhile, the tidal flat area has shrunk to 320 km². All together, these changes resulted in the present morphological features of Qinzhou Bay with a staggered distribution of artificial coast, islands and reefs, and natural coastline, coupled with complex underwater topography and geomorphology. Together with the channel effect of the Longmen Channel where the velocity of the ebb tide is higher than that of the flood tide, the underwater topography of outer Qinzhou Bay is characterized by increased scale and height difference of troughs and ridges, as well as enhanced near-shore deposition.

**Author Contributions:** Writing—original draft, C.C.; Writing—review and editing, F.C. and H.Q.; Investigation Y.Z. and H.L.; all authors reviewed the manuscript. All authors have read and agreed to the published version of the manuscript.

**Funding:** This research was funded by the National Natural Science Foundation of China (Grant No. 42076058, 41930538 and 41406059), the Scientific Research Foundation of Third Institute of Oceanography, MNR (Grant No. 2019006, 2016037 and 2020017), the Province Natural Science Foundation of Fujian (Grant No. 2016J01190) and Special Funds for Scientific Research on Marine Public Causes (Grant No. 201505012-5).

**Institutional Review Board Statement:** Not applicable.

**Informed Consent Statement:** Not applicable.

**Data Availability Statement:** Data is contained within the article. The data presented in this study are available in Figures 3–6 and 8 and Table 1.

**Acknowledgments:** The authors would like to express their sincere thanks to all those who have offered support.

**Conflicts of Interest:** The authors declare no conflict of interest.

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
