# Peer review of "Characteristics of Underwater Topography, Geomorphology and Sediment Source in Qinzhou Bay"

_water, doi:10.3390/w13101392_

Round 1
Reviewer 1 Report
General comments:
In section 4.3, it is not known how you delineated the distribution of sediment types in the study site, presented in Figure 6. Please, supplement the explanation in the methods section.
Also, supplement scale bars in all maps representing Qinzhou Bay.
Moreover, I suggest enriching the introduction for some recent works utilizing underwater acoustics for the determination of anthropogenic impact in different shallow water environments. Consider i.e.:
- Fogarin, S.; Madricardo, F.; Zaggia, L.; Sigovini, M.; Montereale‐Gavazzi, G.; Kruss, A.; Lorenzetti, G.; Manfé, G.; Petrizzo, A.; Molinaroli, E.; Trincardi, F. Tidal inlets in the Anthropocene: geomorphology and benthic habitats of the Chioggia inlet, Venice Lagoon (Italy). Earth Surface Processes and Landforms 2019, 44, 2297–2315, doi:10.1002/esp.4642.
- Janowski, L., Kubacka, M., Pydyn, A., Popek, M., Gajewski, L., 2021. From acoustics to underwater archaeology: Deep investigation of a shallow lake using high‐resolution hydroacoustics—The case of Lake Lednica, Poland. Archaeometry.
- Watson, S.J.; Neil, H.; Ribó, M.; Lamarche, G.; Strachan, L.J.; MacKay, K.; Wilcox, S.; Kane, T.; Orpin, A.; Nodder, S.; Pallentin, A.; Steinmetz, T. What We Do in the Shallows: Natural and Anthropogenic Seafloor Geomorphologies in a Drowned River Valley, New Zealand. Frontiers in Marine Science 2020, 7, doi:10.3389/fmars.2020.579626.
Specific remarks:
line 118: reference needed
line 120-121: hectometer is rather rarely used in the literature. Consider changing it either to m or km.
line 149: supplement the measuring platform - which specific scientific vessel/boat was used?
line 151-152: provide acoustic characteristics of the device, like operational frequency of the measurements, pulse length, etc.
line 166: provide the frequency of SVP measurements
line 171: provide the used positioning and motion reference system
line 173: when sediment samples were collected?
line 188: provide the version of software
line 190: what was the source for tidal corrections? Any reference stations? DGPS RTK corrections?
line 192: provide the version of software
line 196: explain what do you mean by man-machine interaction?
lines 300-301: methodological sentence. You may also refer to some recent scientific works related to automatic seafloor mapping based on multibeam echosounder bathymetry and backscatter data.
I would like to see this paper published after considering the comments mentioned above.
Author Response
Response to Review Comments (Water-1200609)
We are greatly encouraged by the positive comments by the Guest Editor and the Chief Editor in view of our previously submitted manuscript through a peer review. A special thanks to the Guest Editor, the Chief Editor, and two anonymous reviewers for their deep and thorough reviews. We have tried our best to modify the previous submission of this manuscript regarding all the detailed comments and constructive suggestions. Please find below a point-by-point reply relative to all comments raised by the tow anonymous reviewers. To better show how and where the earlier version is thoroughly revised, both the “track changes” and clean version of the manuscript based on the previous submission are provided. All line numbers mentioned in this detailed reply belong to the manuscript with “track changes”.
Reviewer1#:
General comments:
In section 4.3, it is not known how you delineated the distribution of sediment types in the study site, presented in Figure 6. Please, supplement the explanation in the methods section.
Also, supplement scale bars in all maps representing Qinzhou Bay.
Moreover, I suggest enriching the introduction for some recent works utilizing underwater acoustics for the determination of anthropogenic impact in different shallow water environments. Consider i.e.:
- Fogarin, S.; Madricardo, F.; Zaggia, L.; Sigovini, M.; Montereale‐Gavazzi, G.; Kruss, A.; Lorenzetti, G.; Manfé, G.; Petrizzo, A.; Molinaroli, E.; Trincardi, F. Tidal inlets in the Anthropocene: geomorphology and benthic habitats of the Chioggia inlet, Venice Lagoon (Italy). Earth Surface Processes and Landforms. 2019, 44, 2297–2315, doi:10.1002/esp.4642.
- Janowski, L., Kubacka, M., Pydyn, A., Popek, M., Gajewski, L. From acoustics to underwater archaeology: Deep investigation of a shallow lake using high-resolution hydroacoustics-The case of Lake Lednica, Poland. Archaeometry. 2021. 12663.
- Watson, S.J.; Neil, H.; Ribó, M.; Lamarche, G.; Strachan, L.J.; MacKay, K.; Wilcox, S.; Kane, T.; Orpin, A.; Nodder, S.; Pallentin, A.; Steinmetz, T. What We Do in the Shallows: Natural and Anthropogenic Seafloor Geomorphologies in a Drowned River Valley, New Zealand. Frontiers in Marine Science. 2020, 7, doi:10.3389/fmars.2020.579626.
Response: Thank you very much for the reviewer's recognition of our article. Your encouragement is the greatest recognition of our scientific research. We made some revision to the article, including analysis and data presentation. Thank you very much for your help. The major revisions include: (1) we suppled test method of sediment types in the methods section 3.2. (2) We have suppled scale bars in all figures (Fig.1-Fig.6 and Fig.9). (3) Added three references into section 3.1 and section 5.3, number as 36, 37 and 28. (4) Enriched the introduction for some recent works utilizing underwater acoustics in the introduction section 1.
Specific remarks:
1、line 118: reference needed
Response: We have supplied a reference No.31.
2、line 120-121: hectometer is rather rarely used in the literature. Consider changing it either to m or km.
Response: “hm” means hectare in this paper. Hectare is abbreviate to hm.
3、line 149: supplement the measuring platform - which specific scientific vessel/boat was used?
Response: We have supplied the measuring platform a cruise of the Boat Qinyu 26957. See the revision in section 3.1 of the new manuscript with track changes.
4、line 151-152: provide acoustic characteristics of the device, like operational frequency of the measurements, pulse length, etc.
Response: We have supplied operational frequency of the measurements is 208KHz in section 3.1.
5、line 166: provide the frequency of SVP measurements.
Response: We have supplied the frequency of SVP measurements is 30Hz.
6、line 171: provide the used positioning and motion reference system.
Response: We have supplied that DGPS positioning system is used for offshore operation positioning, and the sailing positioning accuracy is better than 1.0m.
7、line 173: when sediment samples were collected?
Response: Sediment samples were collected is in March 2017. We have supplied in this section.
8、line 188: provide the version of software.
Response: The version of Hypack software is v18.1.
9、line 190: what was the source for tidal corrections? Any reference stations? DGPS RTK corrections?
Response: During the bathymetric survey, the GXCORS non-tidal station was used to synchronously calibrate the tide level. In this study, the planar control solution of 6 control points in the test area was carried out by using GAMIT software and 5 IGS stations (BJFS, Hyde, Pimo, Shao, TWTF). We have supplied in section 3.2.
10、line 192: provide the version of software.
Response: We have supplied the version of ArcGIS software is v9.1.
11、line 196: explain what do you mean by man-machine interaction?
Response: Man-machine interaction means that the instrument displays the real-time seabed depth data graphically, identifies the acquired depth data, and provides qualified depth profile lines for manual evaluation. The processing personnel can interpret and correct the false data in real time. We have supplied the previous sentence.
12、lines 300-301: methodological sentence. You may also refer to some recent scientific works related to automatic seafloor mapping based on multibeam echosounder bathymetry and backscatter data.
Response: Many thanks to the reviewer for this suggestion. We have modified methodological sentence refer to multibeam echosounder bathymetry and backscatter data. In my future work, I will try my best to refer to the latest scientific research work related to automatic seafloor mapping.
I would like to see this paper published after considering the comments mentioned above.

Reviewer 2 Report
The article is interesting and well written.
I have a little attention to the research methodsL
How the samples were taken? What instrument was used?
Kind regards
Author Response
Response to Review Comments (Water-1200609)
We are greatly encouraged by the positive comments by the Guest Editor and the Chief Editor in view of our previously submitted manuscript through a peer review. A special thanks to the Guest Editor, the Chief Editor, and two anonymous reviewers for their deep and thorough reviews. We have tried our best to modify the previous submission of this manuscript regarding all the detailed comments and constructive suggestions. Please find below a point-by-point reply relative to all comments raised by the tow anonymous reviewers. To better show how and where the earlier version is thoroughly revised, both the “track changes” and clean version of the manuscript based on the previous submission are provided. All line numbers mentioned in this detailed reply belong to the manuscript with “track changes”.
Reviewer2
The article is interesting and well written. I have a little attention to the research methods. How the samples were taken? What instrument was used?
Response: Thank you very much for the reviewer's recognition of our article. Your encouragement is the greatest recognition of our scientific research. We made some revision to the article, including analysis and data presentation. Thank you very much for your help.
We grabbed to obtain surface sediment samples and collected columnar sediment samples with gravity tube. We have supplied the previous sentence in section 3.1.

This manuscript is a resubmission of an earlier submission. The following is a list of the peer review reports and author responses from that submission.
Round 1
Reviewer 1 Report
I see that you did all the necesssary corrections that I had noted on the initial version of the manuscript.. The new form of the paper is satisfying for me. Well done.